# Entropy and mutual information in models of deep neural networks

**Marylou Gabrié**[*,1], **Andre Manoel**[2,3], **Clément Luneau**[4], **Jean Barbier**[1,4,5], **Nicolas Macris**[4], **Florent Krzakala**[1,6,7] **and Lenka Zdeborová**[3,6]

[1]Laboratoire de Physique Statistique, École Normale Supérieure, PSL University
[2]Parietal Team, INRIA, CEA, Université Paris-Saclay & Owkin Inc., New York
[3]Institut de Physique Théorique, CEA, CNRS, Université Paris-Saclay
[4]Laboratoire de Théorie des Communications, École Polytechnique Fédérale de Lausanne
[5]International Center for Theoretical Physics, Trieste, Italy
[6]Department of Mathematics, Duke University, Durham NC
[7]Sorbonne Universités & LightOn Inc., Paris

## Abstract

We examine a class of stochastic deep learning models with a tractable method to compute information-theoretic quantities. Our contributions are three-fold: (i) We show how entropies and mutual informations can be derived from heuristic statistical physics methods, under the assumption that weight matrices are independent and orthogonally-invariant. (ii) We extend particular cases in which this result is known to be rigorously exact by providing a proof for two-layers networks with Gaussian random weights, using the recently introduced adaptive interpolation method. (iii) We propose an experiment framework with generative models of synthetic datasets, on which we train deep neural networks with a weight constraint designed so that the assumption in (i) is verified during learning. We study the behavior of entropies and mutual informations throughout learning and conclude that, in the proposed setting, the relationship between compression and generalization remains elusive.

The successes of deep learning methods have spurred efforts towards quantitative modeling of the performance of deep neural networks. In particular, an information-theoretic approach linking generalization capabilities to compression has been receiving increasing interest. The intuition behind the study of mutual informations in latent variable models dates back to the information bottleneck (IB) theory of [1]. Although recently reformulated in the context of deep learning [2], verifying its relevance in practice requires the computation of mutual informations for high-dimensional variables, a notoriously hard problem. Thus, pioneering works in this direction focused either on small network models with discrete (continuous, eventually binned) activations [3], or on linear networks [4, 5].

In the present paper we follow a different direction, and build on recent results from statistical physics [6, 7] and information theory [8, 9] to propose, in Section 1, a formula to compute information-theoretic quantities for a class of deep neural network models. The models we approach, described in Section 2, are non-linear feed-forward neural networks trained on synthetic datasets with constrained weights. Such networks capture some of the key properties of the deep learning setting that are usually difficult to include in tractable frameworks: non-linearities, arbitrary large width and depth, and correlations in the input data. We demonstrate the proposed method in a series of numerical experiments in Section 3. First observations suggest a rather complex picture, where the role of compression in the generalization ability of deep neural networks is yet to be elucidated.

---

[*]Corresponding author: marylou.gabrie@ens.fr

# 1 Multi-layer model and main theoretical results

**A stochastic multi-layer model—** We consider a model of multi-layer stochastic feed-forward neural network where each element $x_i$ of the input layer $\boldsymbol{x} \in \mathbb{R}^{n_0}$ is distributed independently as $P_0(x_i)$, while hidden units $t_{\ell,i}$ at each successive layer $\boldsymbol{t}_\ell \in \mathbb{R}^{n_\ell}$ (vectors are column vectors) come from $P_\ell(t_{\ell,i}|\boldsymbol{W}_{\ell,i}^\mathsf{T}\boldsymbol{t}_{\ell-1})$, with $\boldsymbol{t}_0 \equiv \boldsymbol{x}$ and $\boldsymbol{W}_{\ell,i}$ denoting the $i$-th row of the matrix of weights $W_\ell \in \mathbb{R}^{n_\ell \times n_{\ell-1}}$. In other words

$$t_{0,i} \equiv x_i \sim P_0(\cdot), \quad t_{1,i} \sim P_1(\cdot|\boldsymbol{W}_{1,i}^\mathsf{T}\boldsymbol{x}), \quad \dots \quad t_{L,i} \sim P_L(\cdot|\boldsymbol{W}_{L,i}^\mathsf{T}\boldsymbol{t}_{L-1}), \tag{1}$$

given a set of weight matrices $\{W_\ell\}_{\ell=1}^L$ and distributions $\{P_\ell\}_{\ell=1}^L$ which encode possible non-linearities and stochastic noise applied to the hidden layer variables, and $P_0$ that generates the visible variables. In particular, for a non-linearity $t_{\ell,i} = \varphi_\ell(h, \xi_{\ell,i})$, where $\xi_{\ell,i} \sim P_\xi(\cdot)$ is the stochastic noise (independent for each $i$), we have $P_\ell(t_{\ell,i}|\boldsymbol{W}_{\ell,i}^\mathsf{T}\boldsymbol{t}_{\ell-1}) = \int dP_\xi(\xi_{\ell,i})\,\delta\big(t_{\ell,i}-\varphi_\ell(\boldsymbol{W}_{\ell,i}^\mathsf{T}\boldsymbol{t}_{\ell-1}, \xi_{\ell,i})\big)$. Model (1) thus describes a Markov chain which we denote by $\boldsymbol{X} \to \boldsymbol{T}_1 \to \boldsymbol{T}_2 \to \cdots \to \boldsymbol{T}_L$, with $\boldsymbol{T}_\ell = \varphi_\ell(W_\ell \boldsymbol{T}_{\ell-1}, \boldsymbol{\xi}_\ell)$, $\boldsymbol{\xi}_\ell = \{\xi_{\ell,i}\}_{i=1}^{n_\ell}$, and the activation function $\varphi_\ell$ applied componentwise.

**Replica formula—** We shall work in the asymptotic high-dimensional statistics regime where all $\tilde{\alpha}_\ell \equiv n_\ell/n_0$ are of order one while $n_0 \to \infty$, and make the important assumption that all matrices $W_\ell$ are orthogonally-invariant random matrices independent from each other; in other words, each matrix $W_\ell \in \mathbb{R}^{n_\ell \times n_{\ell-1}}$ can be decomposed as a product of three matrices, $W_\ell = U_\ell S_\ell V_\ell$, where $U_\ell \in \mathrm{O}(n_\ell)$ and $V_\ell \in \mathrm{O}(n_{\ell-1})$ are independently sampled from the Haar measure, and $S_\ell$ is a diagonal matrix of singular values. The main technical tool we use is a formula for the entropies of the hidden variables, $H(\boldsymbol{T}_\ell) = -\mathbb{E}_{\boldsymbol{T}_\ell} \ln P_{\boldsymbol{T}_\ell}(\boldsymbol{t}_\ell)$, and the mutual information between adjacent layers $I(\boldsymbol{T}_\ell; \boldsymbol{T}_{\ell-1}) = H(\boldsymbol{T}_\ell) + \mathbb{E}_{\boldsymbol{T}_\ell, \boldsymbol{T}_{\ell-1}} \ln P_{\boldsymbol{T}_\ell|\boldsymbol{T}_{\ell-1}}(\boldsymbol{t}_\ell|\boldsymbol{t}_{\ell-1})$, based on the heuristic replica method [10, 11, 6, 7]:

**Claim 1** (Replica formula). *Assume model (1) with $L$ layers in the high-dimensional limit with componentwise activation functions and weight matrices generated from the ensemble described above, and denote by $\lambda_{W_k}$ the eigenvalues of $W_k^\mathsf{T} W_k$. Then for any $\ell \in \{1, \dots, L\}$ the normalized entropy of $\boldsymbol{T}_\ell$ is given by the minimum among all stationary points of the replica potential:*

$$\lim_{n_0 \to \infty} \frac{1}{n_0} H(\boldsymbol{T}_\ell) = \min_{\boldsymbol{A}, \boldsymbol{V}, \tilde{\boldsymbol{A}}, \tilde{\boldsymbol{V}}} \phi_\ell(\boldsymbol{A}, \boldsymbol{V}, \tilde{\boldsymbol{A}}, \tilde{\boldsymbol{V}}), \tag{2}$$

*which depends on $\ell$-dimensional vectors $\boldsymbol{A}, \boldsymbol{V}, \tilde{\boldsymbol{A}}, \tilde{\boldsymbol{V}}$, and is written in terms of mutual information $I$ and conditional entropies $H$ of scalar variables as*

$$\phi_\ell(\boldsymbol{A}, \boldsymbol{V}, \tilde{\boldsymbol{A}}, \tilde{\boldsymbol{V}}) = I\Big(t_0; t_0 + \frac{\xi_0}{\sqrt{\tilde{A}_1}}\Big) - \frac{1}{2}\sum_{k=1}^\ell \tilde{\alpha}_{k-1}\big[\tilde{A}_k V_k + \alpha_k A_k \tilde{V}_k - F_{W_k}(A_k V_k)\big]$$

$$+ \sum_{k=1}^{\ell-1} \tilde{\alpha}_k \Big[H(t_k|\xi_k; \tilde{A}_{k+1}, \tilde{V}_k, \tilde{\rho}_k) - \frac{1}{2}\log(2\pi e \tilde{A}_{k+1}^{-1})\Big] + \tilde{\alpha}_\ell H(t_\ell|\xi_\ell; \tilde{V}_\ell, \tilde{\rho}_\ell), \tag{3}$$

*where $\alpha_k = n_k/n_{k-1}$, $\tilde{\alpha}_k = n_k/n_0$, $\rho_k = \int dP_{k-1}(t)\,t^2$, $\tilde{\rho}_k = (\mathbb{E}_{\lambda_{W_k}} \lambda_{W_k})\rho_k/\alpha_k$, and $\xi_k \sim \mathcal{N}(0,1)$ for $k = 0, \dots, \ell$. In the computation of the conditional entropies in (3), the scalar $t_k$-variables are generated from $P(t_0) = P_0(t_0)$ and*

$$P(t_k|\xi_k; A, V, \rho) = \mathbb{E}_{\tilde{\xi}, \tilde{z}}\, P_k(t_k + \tilde{\xi}/\sqrt{A}|\sqrt{\rho - V}\xi_k + \sqrt{V}\tilde{z}), \quad k = 1, \dots, \ell-1, \tag{4}$$

$$P(t_\ell|\xi_\ell; V, \rho) = \mathbb{E}_{\tilde{z}}\, P_\ell(t_\ell|\sqrt{\rho - V}\xi_\ell + \sqrt{V}\tilde{z}), \tag{5}$$

*where $\tilde{\xi}$ and $\tilde{z}$ are independent $\mathcal{N}(0,1)$ random variables. Finally, the function $F_{W_k}(x)$ depends on the distribution of the eigenvalues $\lambda_{W_\ell}$ following*

$$F_{W_k}(x) = \min_{\theta \in \mathbb{R}} \big\{2\alpha_k \theta + (\alpha_k - 1)\ln(1 - \theta) + \mathbb{E}_{\lambda_{W_k}} \ln[x\lambda_{W_k} + (1 - \theta)(1 - \alpha_k \theta)]\big\}. \tag{6}$$

The computation of the entropy in the large dimensional limit, a computationally difficult task, has thus been reduced to an extremization of a function of $4\ell$ variables, that requires evaluating single or bidimensional integrals. This extremization can be done efficiently by means of a fixed-point iteration starting from different initial conditions, as detailed in the Supplementary Material [12]. Moreover, a

user-friendly Python package is provided [13], which performs the computation for different choices of prior $P_0$, activations $\varphi_\ell$ and spectra $\lambda_{W_\ell}$. Finally, the mutual information between successive layers $I(\boldsymbol{T}_\ell; \boldsymbol{T}_{\ell-1})$ can be obtained from the entropy following the evaluation of an additional bidimensional integral, see Section 1.6.1 of the Supplementary Material [12].

Our approach in the derivation of (3) builds on recent progresses in statistical estimation and information theory for generalized linear models following the application of methods from statistical physics of disordered systems [10, 11] in communication [14], statistics [15] and machine learning problems [16, 17]. In particular, we use advanced mean field theory [18] and the heuristic replica method [10, 6], along with its recent extension to multi-layer estimation [7, 8], in order to derive the above formula (3). This derivation is lengthy and thus given in the Supplementary Material [12]. In a related contribution, Reeves [9] proposed a formula for the mutual information in the multi-layer setting, using heuristic information-theoretic arguments. As ours, it exhibits layer-wise additivity, and the two formulas are conjectured to be equivalent.

**Rigorous statement—** We recall the assumptions under which the replica formula of Claim 1 is conjectured to be exact: *(i) weight matrices are drawn from an ensemble of random orthogonally-invariant matrices, (ii) matrices at different layers are statistically independent and (iii) layers have a large dimension and respective sizes of adjacent layers are such that weight matrices have aspect ratios $\{\alpha_k, \tilde{\alpha}_k\}_{k=1}^{\ell}$ of order one.* While we could not prove the replica prediction in full generality, we stress that it comes with multiple credentials: (i) for Gaussian prior $P_0$ and Gaussian distributions $P_\ell$, it corresponds to the exact analytical solution when weight matrices are independent of each other (see Section 1.6.2 of the Supplementary Material [12]). (ii) In the single-layer case with a Gaussian weight matrix, it reduces to formula (6) in the Supplementary Material [12], which has been recently rigorously proven for (almost) all activation functions $\varphi$ [19]. (iii) In the case of Gaussian distributions $P_\ell$, it has also been proven for a large ensemble of random matrices [20] and (iv) it is consistent with all the results of the AMP [21, 22, 23] and VAMP [24] algorithms, and their multi-layer versions [7, 8], known to perform well for these estimation problems.

In order to go beyond results for the single-layer problem and heuristic arguments, we prove Claim 1 for the more involved multi-layer case, assuming Gaussian i.i.d. matrices and two non-linear layers:

**Theorem 1** (Two-layer Gaussian replica formula). *Suppose $(H1)$ the input units distribution $P_0$ is separable and has bounded support; $(H2)$ the activations $\varphi_1$ and $\varphi_2$ corresponding to $P_1(t_{1,i}|\boldsymbol{W}_{1,i}^\mathsf{T}\boldsymbol{x})$ and $P_2(t_{2,i}|\boldsymbol{W}_{2,i}^\mathsf{T}\boldsymbol{t}_1)$ are bounded $\mathcal{C}^2$ with bounded first and second derivatives w.r.t their first argument; and $(H3)$ the weight matrices $W_1, W_2$ have Gaussian i.i.d. entries. Then for model (1) with two layers $L = 2$ the high-dimensional limit of the entropy verifies Claim 1.*

The theorem, that closes the conjecture presented in [7], is proven using the adaptive interpolation method of [25, 19] in a multi-layer setting, as first developed in [26]. The lengthy proof, presented in details in the Supplementary Material [12], is of independent interest and adds further credentials to the replica formula, as well as offers a clear direction to further developments. Note that, following the same approximation arguments as in [19] where the proof is given for the single-layer case, the hypothesis $(H1)$ can be relaxed to the existence of the second moment of the prior, $(H2)$ can be dropped and $(H3)$ extended to matrices with i.i.d. entries of zero mean, $O(1/n_0)$ variance and finite third moment.

## 2 Tractable models for deep learning

The multi-layer model presented above can be leveraged to simulate two prototypical settings of deep supervised learning on synthetic datasets amenable to the replica tractable computation of entropies and mutual informations.

The first scenario is the so-called *teacher-student* (see Figure 1, left). Here, we assume that the input $\boldsymbol{x}$ is distributed according to a *separable* prior distribution $P_X(\boldsymbol{x}) = \prod_i P_0(x_i)$, factorized in the components of $\boldsymbol{x}$, and the corresponding label $\boldsymbol{y}$ is given by applying a mapping $\boldsymbol{x} \to \boldsymbol{y}$, called *the teacher*. After generating a train and test set in this manner, we perform the training of a deep neural network, *the student*, on the synthetic dataset. In this case, the data themselves have a simple structure given by $P_0$.

In constrast, the second scenario allows *generative models* (see Figure 1, right) that create more structure, and that are reminiscent of the *generative-recognition* pair of models of a Variational

Autoencoder (VAE). A code vector $\boldsymbol{y}$ is sampled from a separable prior distribution $P_Y(\boldsymbol{y}) = \prod_i P_0(y_i)$ and a corresponding data point $\boldsymbol{x}$ is generated by a possibly stochastic neural network, *the generative model*. This setting allows to create input data $\boldsymbol{x}$ featuring correlations, differently from the teacher-student scenario. The studied supervised learning task then consists in training a deep neural net, *the recognition model*, to recover the code $\boldsymbol{y}$ from $\boldsymbol{x}$.

In both cases, the chain going from $\boldsymbol{X}$ to any later layer is a Markov chain in the form of (1). In the first scenario, model (1) directly maps to the student network. In the second scenario however, model (1) actually maps to the feed-forward combination of the generative model followed by the recognition model. This shift is necessary to verify the assumption that the starting point (now given by $\boldsymbol{Y}$) has a separable distribution. In particular, it generates correlated input data $\boldsymbol{X}$ while still allowing for the computation of the entropy of any $\boldsymbol{T}_\ell$.

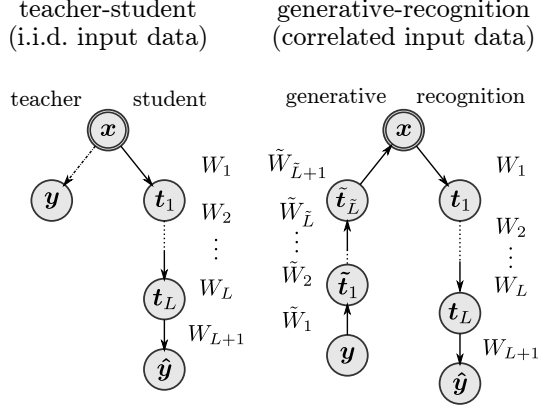

Figure 1: Two models of synthetic data

At the start of a neural network training, weight matrices initialized as i.i.d. Gaussian random matrices satisfy the necessary assumptions of the formula of Claim 1. In their singular value decomposition

$$W_\ell = U_\ell S_\ell V_\ell \tag{7}$$

the matrices $U_\ell \in \mathrm{O}(n_\ell)$ and $V_\ell \in \mathrm{O}(n_{\ell-1})$, are typical independent samples from the Haar measure across all layers. To make sure weight matrices remain close enough to independent during learning, we define a custom weight constraint which consists in keeping $U_\ell$ and $V_\ell$ fixed while only the matrix $S_\ell$, constrained to be diagonal, is updated. The number of parameters is thus reduced from $n_\ell \times n_{\ell-1}$ to $\min(n_\ell, n_{\ell-1})$. We refer to layers following this weight constraint as USV-layers. For the replica formula of Claim 1 to be correct, the matrices $S_\ell$ from different layers should furthermore remain uncorrelated during the learning. In Section 3, we consider the training of linear networks for which information-theoretic quantities can be computed analytically, and confirm numerically that with USV-layers the replica predicted entropy is correct at all times. In the following, we assume that is also the case for non-linear networks.

In Section 3.2 of the Supplementary Material [12] we train a neural network with USV-layers on a simple real-world dataset (MNIST), showing that these layers can learn to represent complex functions despite their restriction. We further note that such a product decomposition is reminiscent of a series of works on adaptative structured efficient linear layers (SELLs and ACDC) [27, 28] motivated this time by speed gains, where only diagonal matrices are learned (in these works the matrices $U_\ell$ and $V_\ell$ are chosen instead as permutations of Fourier or Hadamard matrices, so that the matrix multiplication can be replaced by fast transforms). In Section 3, we discuss learning experiments with USV-layers on synthetic datasets.

While we have defined model (1) as a stochastic model, traditional feed forward neural networks are deterministic. In the numerical experiments of Section 3, we train and test networks without injecting noise, and only assume a noise model in the computation of information-theoretic quantities. Indeed, for continuous variables the presence of noise is necessary for mutual informations to remain finite (see discussion of Appendix C in [5]). We assume at layer $\ell$ an additive white Gaussian noise of small amplitude just before passing through its activation function to obtain $H(\boldsymbol{T}_\ell)$ and $I(\boldsymbol{T}_\ell; \boldsymbol{T}_{\ell-1})$, while keeping the mapping $\boldsymbol{X} \to \boldsymbol{T}_{\ell-1}$ deterministic. This choice attempts to stay as close as possible to the deterministic neural network, but remains inevitably somewhat arbitrary (see again discussion of Appendix C in [5]).

**Other related works—** The strategy of studying neural networks models, with random weight matrices and/or random data, using methods originated in statistical physics heuristics, such as the replica and the cavity methods [10] has a long history. Before the deep learning era, this approach led to pioneering results in learning for the Hopfield model [29] and for the random perceptron [30, 31, 16, 17].

Recently, the successes of deep learning along with the disqualifying complexity of studying real world problems have sparked a revived interest in the direction of random weight matrices. Recent results –without exhaustivity– were obtained on the spectrum of the Gram matrix at each layer using random matrix theory [32, 33], on expressivity of deep neural networks [34], on the dynamics of propagation and learning [35, 36, 37, 38], on the high-dimensional non-convex landscape where the learning takes place [39], or on the universal random Gaussian neural nets of [40].

The information bottleneck theory [1] applied to neural networks consists in computing the mutual information between the data and the learned hidden representations on the one hand, and between labels and again hidden learned representations on the other hand [2, 3]. A successful training should maximize the information with respect to the labels and simultaneously minimize the information with respect to the input data, preventing overfitting and leading to a good generalization. While this intuition suggests new learning algorithms and regularizers [41, 42, 43, 44, 45, 46, 47], we can also hypothesize that this mechanism is already at play in a priori unrelated commonly used optimization methods, such as the simple stochastic gradient descent (SGD). It was first tested in practice by [3] on very small neural networks, to allow the entropy to be estimated by binning of the hidden neurons activities. Afterwards, the authors of [5] reproduced the results of [3] on small networks using the continuous entropy estimator of [45], but found that the overall behavior of mutual information during learning is greatly affected when changing the nature of non-linearities. Additionally, they investigate the training of larger linear networks on i.i.d. normally distributed inputs where entropies at each hidden layer can be computed analytically for an additive Gaussian noise. The strategy proposed in the present paper allows us to evaluate entropies and mutual informations in non-linear networks larger than in [5, 3].

# 3   Numerical experiments

We present a series of experiments both aiming at further validating the replica estimator and leveraging its power in noteworthy applications. A first application presented in the paragraph 3.1 consists in using the replica formula in settings where it is proven to be rigorously exact as a basis of comparison for other entropy estimators. The same experiment also contributes to the discussion of the information bottleneck theory for neural networks by showing how, without any learning, information-theoretic quantities have different behaviors for different non-linearities. In the following paragraph 3.2, we validate the accuracy of the replica formula in a learning experiment with USV-layers —where it is not proven to be exact — by considering the case of linear networks for which information-theoretic quantities can be otherwise computed in closed-form. We finally consider in the paragraph 3.3, a second application testing the information bottleneck theory for large non-linear networks. To this aim, we use the replica estimator to study compression effects during learning.

**3.1 Estimators and activation comparisons—** Two non-parametric estimators have already been considered by [5] to compute entropies and/or mutual informations during learning. The kernel-density approach of Kolchinsky et. al. [45] consists in fitting a mixture of Gaussians (MoG) to samples of the variable of interest and subsequently compute an upper bound on the entropy of the MoG [48]. The method of Kraskov et al. [49] uses nearest neighbor distances between samples to directly build an estimate of the entropy. Both methods require the computation of the matrix of distances between samples. Recently, [46] proposed a new non-parametric estimator for mutual informations which involves the optimization of a neural network to tighten a bound. It is unfortunately computationally hard to test how these estimators behave in high dimension as even for a known distribution the computation of the entropy is intractable (#P-complete) in most cases. However the replica method proposed here is a valuable point of comparison for cases where it is rigorously exact.

In the first numerical experiment we place ourselves in the setting of Theorem 1: a 2-layer network with i.i.d weight matrices, where the formula of Claim 1 is thus rigorously exact in the limit of large networks, and we compare the replica results with the non-parametric estimators of [45] and [49]. Note that the requirement for smooth activations $(H2)$ of Theorem 1 can be relaxed (see discussion below the Theorem). Additionally, non-smooth functions can be approximated arbitrarily closely by smooth functions with equal information-theoretic quantities, up to numerical precision.

We consider a neural network with layers of equal size $n = 1000$ that we denote: $\boldsymbol{X} \rightarrow \boldsymbol{T}_1 \rightarrow \boldsymbol{T}_2$. The input variable components are i.i.d. Gaussian with mean 0 and variance 1. The weight matrices

entries are also i.i.d. Gaussian with mean 0. Their standard-deviation is rescaled by a factor $1/\sqrt{n}$ and then multiplied by a coefficient $\sigma$ varying between 0.1 and 10, i.e. around the recommended value for training initialization. To compute entropies, we consider noisy versions of the latent variables where an additive white Gaussian noise of very small variance ($\sigma_{\text{noise}}^2 = 10^{-5}$) is added right before the activation function, $\boldsymbol{T}_1 = f(W_1\boldsymbol{X} + \boldsymbol{\epsilon}_1)$ and $\boldsymbol{T}_2 = f(W_2 f(W_1\boldsymbol{X}) + \boldsymbol{\epsilon}_2)$ with $\boldsymbol{\epsilon}_{1,2} \sim \mathcal{N}(0, \sigma_{\text{noise}}^2 I_n)$, which is also done in the remaining experiments to guarantee the mutual informations to remain finite. The non-parametric estimators [45, 49] were evaluated using 1000 samples, as the cost of computing pairwise distances is significant in such high dimension and we checked that the entropy estimate is stable over independent draws of a sample of such a size (error bars smaller than marker size). On Figure 2, we compare the different estimates of $H(\boldsymbol{T}_1)$ and $H(\boldsymbol{T}_2)$ for different activation functions: linear, hardtanh or ReLU. The hardtanh activation is a piecewise linear approximation of the tanh, $\text{hardtanh}(x) = -1$ for $x < -1$, $x$ for $-1 < x < 1$, and 1 for $x > 1$, for which the integrals in the replica formula can be evaluated faster than for the tanh.

In the linear and hardtanh case, the non-parametric methods are following the tendency of the replica estimate when $\sigma$ is varied, but appear to systematically over-estimate the entropy. For linear networks with Gaussian inputs and additive Gaussian noise, every layer is also a multivariate Gaussian and therefore entropies can be directly computed in closed form (*exact* in the plot legend). When using the Kolchinsky estimate in the linear case we also check the consistency of two strategies, either fitting the MoG to the noisy sample or fitting the MoG to the deterministic part of the $\boldsymbol{T}_\ell$ and augment the resulting variance with $\sigma_{\text{noise}}^2$, as done in [45] (*Kolchinsky et al. parametric* in the plot legend). In the network with hardtanh non-linearities, we check that for small weight values, the entropies are the same as in a linear network with same weights (*linear approx* in the plot legend, computed using the exact analytical result for linear networks and therefore plotted in a similar color to *exact*). Lastly, in the case of the ReLU-ReLU network, we note that non-parametric methods are predicting an entropy increasing as the one of a linear network with identical weights, whereas the replica computation reflects its knowledge of the cut-off and accurately features a slope equal to half of the linear network entropy (*1/2 linear approx* in the plot legend). While non-parametric estimators are invaluable tools able to approximate entropies from the mere knowledge of samples, they inevitably introduce estimation errors. The replica method is taking the opposite view. While being restricted to a class of models, it can leverage its knowledge of the neural network structure to provide a reliable estimate. To our knowledge, there is no other entropy estimator able to incorporate such information about the underlying multi-layer model.

Beyond informing about estimators accuracy, this experiment also unveils a simple but possibly important distinction between activation functions. For the hardtanh activation, as the random weights magnitude increases, the entropies decrease after reaching a maximum, whereas they only increase for the unbounded activation functions we consider – even for the single-side saturating ReLU. This loss of information for bounded activations was also observed by [5], where entropies were computed by discretizing the output as a single neuron with bins of equal size. In this setting, as the tanh activation starts to saturate for large inputs, the extreme bins (at $-1$ and 1) concentrate more and more probability mass, which explains the information loss. Here we confirm that the phenomenon is also observed when computing the entropy of the hardtanh (without binning and with small noise injected before the non-linearity). We check via the replica formula that the same phenomenology arises for the mutual informations $I(\boldsymbol{X}; \boldsymbol{T}_\ell)$ (see Section 3.1 of the Supplementary Material [12]).

**3.2 Learning experiments with linear networks—** In the following, and in Section 3.3 of the Supplementary Material [12], we discuss training experiments of different instances of the deep learning models defined in Section 2. We seek to study the simplest possible training strategies achieving good generalization. Hence for all experiments we use plain stochastic gradient descent (SGD) with constant learning rates, without momentum and without any explicit form of regularization. The sizes of the training and testing sets are taken equal and scale typically as a few hundreds times the size of the input layer. Unless otherwise stated, plots correspond to single runs, yet we checked over a few repetitions that outcomes of independent runs lead to identical qualitative behaviors. The values of mutual informations $I(\boldsymbol{X}; \boldsymbol{T}_\ell)$ are computed by considering noisy versions of the latent variables where an additive white Gaussian noise of very small variance ($\sigma_{\text{noise}}^2 = 10^{-5}$) is added right before the activation function, as in the previous experiment. This noise is neither present at training time, where it could act as a regularizer, nor at testing time. Given the noise is only assumed at the last layer, the second to last layer is a deterministic mapping of the input variable; hence the replica formula yielding mutual informations between adjacent layers gives us directly

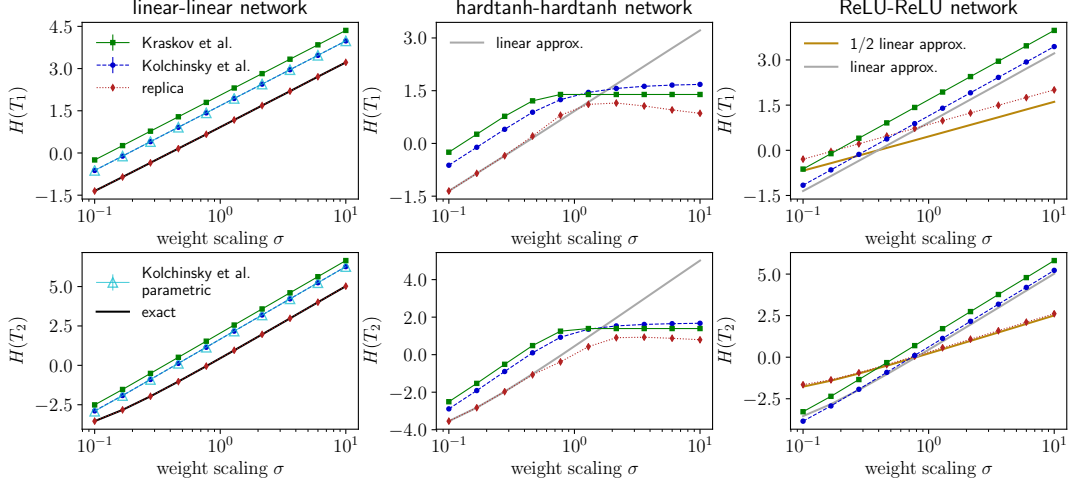

Figure 2: Entropy of latent variables in stochastic networks $\boldsymbol{X} \to \boldsymbol{T}_1 \to \boldsymbol{T}_2$, with equally sized layers $n = 1000$, inputs drawn from $\mathcal{N}(0, I_n)$, weights from $\mathcal{N}(0, \sigma^2 I_{n^2}/n)$, as a function of the weight scaling parameter $\sigma$. An additive white Gaussian noise $\mathcal{N}(0, 10^{-5}I_n)$ is added inside the non-linearity. Left column: linear network. Center column: hardtanh-hardtanh network. Right column: ReLU-ReLU network.

$I(\boldsymbol{T}_\ell; \boldsymbol{T}_{\ell-1}) = H(\boldsymbol{T}_\ell) - H(\boldsymbol{T}_\ell|\boldsymbol{T}_{\ell-1}) = H(\boldsymbol{T}_\ell) - H(\boldsymbol{T}_\ell|\boldsymbol{X}) = I(\boldsymbol{T}_\ell; \boldsymbol{X})$. We provide a second Python package [50] to implement in Keras learning experiments on synthetic datasets, using USV-layers and interfacing the first Python package [13] for replica computations.

To start with we consider the training of a linear network in the teacher-student scenario. The teacher has also to be linear to be learnable: we consider a simple single-layer network with additive white Gaussian noise, $\boldsymbol{Y} = \tilde{W}_{\text{teach}}\boldsymbol{X} + \boldsymbol{\epsilon}$, with input $\boldsymbol{x} \sim \mathcal{N}(0, I_n)$ of size $n$, teacher matrix $\tilde{W}_{\text{teach}}$ i.i.d. normally distributed as $\mathcal{N}(0, 1/n)$, noise $\boldsymbol{\epsilon} \sim \mathcal{N}(0, 0.01I_n)$, and output of size $n_Y = 4$. We train a student network of three USV-layers, plus one fully connected unconstrained layer $\boldsymbol{X} \to \boldsymbol{T}_1 \to \boldsymbol{T}_2 \to \boldsymbol{T}_3 \to \hat{\boldsymbol{Y}}$ on the regression task, using plain SGD for the MSE loss $(\hat{\boldsymbol{Y}} - \boldsymbol{Y})^2$. We recall that in the USV-layers (7) only the diagonal matrix is updated during learning. On the left panel of Figure 3, we report the learning curve and the mutual informations between the hidden layers and the input in the case where all layers but outputs have size $n = 1500$. Again this linear setting is analytically tractable and does not require the replica formula, a similar situation was studied in [5]. In agreement with their observations, we find that the mutual informations $I(\boldsymbol{X}; \boldsymbol{T}_\ell)$ keep on increasing throughout the learning, without compromising the generalization ability of the student. Now, we also use this linear setting to demonstrate (i) that the replica formula remains correct throughout the learning of the USV-layers and (ii) that the replica method gets closer and closer to the exact result in the limit of large networks, as theoretically predicted (2). To this aim, we repeat the experiment for $n$ varying between 100 and 1500, and report the maximum and the mean value of the squared error on the estimation of the $I(\boldsymbol{X}; \boldsymbol{T}_\ell)$ over all epochs of 5 independent training runs. We find that even if errors tend to increase with the number of layers, they remain objectively very small and decrease drastically as the size of the layers increases.

**3.3 Learning experiments with deep non-linear networks—** Finally, we apply the replica formula to estimate mutual informations during the training of non-linear networks on correlated input data.

We consider a simple single layer generative model $\boldsymbol{X} = \tilde{W}_{\text{gen}}\boldsymbol{Y} + \boldsymbol{\epsilon}$ with normally distributed code $\boldsymbol{Y} \sim \mathcal{N}(0, I_{n_Y})$ of size $n_Y = 100$, data of size $n_X = 500$ generated with matrix $\tilde{W}_{\text{gen}}$ i.i.d. normally distributed as $\mathcal{N}(0, 1/n_Y)$ and noise $\boldsymbol{\epsilon} \sim \mathcal{N}(0, 0.01I_{n_X})$. We then train a recognition model to solve the binary classification problem of recovering the label $y = \text{sign}(Y_1)$, the sign of the first neuron in $\boldsymbol{Y}$, using plain SGD but this time to minimize the cross-entropy loss. Note that the rest of the initial code $(Y_2, ..Y_{n_Y})$ acts as noise/nuisance with respect to the learning task. We compare two 5-layers recognition models with 4 USV- layers plus one unconstrained, of sizes 500-1000-500-250-100-2, and activations either linear-ReLU-linear-ReLU-softmax (top row of Figure 4) or linear-hardtanh-linear-hardtanh-softmax (bottom row). Because USV-layers only feature $O(n)$ parameters instead

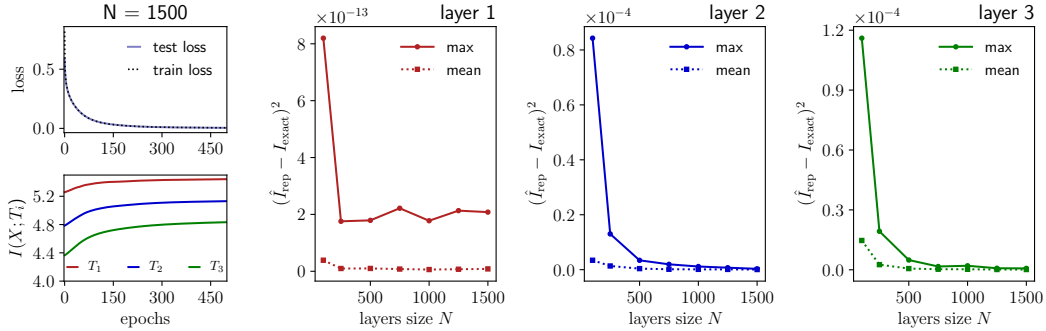

Figure 3: Training of a 4-layer linear student of varying size on a regression task generated by a linear teacher of output size $n_{\boldsymbol{Y}} = 4$. Upper-left: MSE loss on the training and testing sets during training by plain SGD for layers of size $n = 1500$. Best training loss is 0.004735, best testing loss is 0.004789. Lower-left: Corresponding mutual information evolution between hidden layers and input. Center-left, center-right, right: maximum and squared error of the replica estimation of the mutual information as a function of layers size $n$, over the course of 5 independent trainings for each value of $n$ for the first, second and third hidden layer.

of $O(n^2)$ we observe that they require more iterations to train in general. In the case of the ReLU network, adding interleaved linear layers was key to successful training with 2 non-linearities, which explains the somewhat unusual architecture proposed. For the recognition model using hardtanh, this was actually not an issue (see Supplementary Material [12] for an experiment using only hardtanh activations), however, we consider a similar architecture for fair comparison. We discuss further the ability of learning of USV-layers in the Supplementary Material [12].

This experiment is reminiscent of the setting of [3], yet now tractable for networks of larger sizes. For both types of non-linearities we observe that the mutual information between the input and all hidden layers decrease during the learning, except for the very beginning of training where we can sometimes observe a short phase of increase (see zoom in insets). For the hardtanh layers this phase is longer and the initial increase of noticeable amplitude.

In this particular experiment, the claim of [3] that compression can occur during training even with non double-saturated activation seems corroborated (a phenomenon that was not observed by [5]). Yet we do not observe that the compression is more pronounced in deeper layers and its link to generalization remains elusive. For instance, we do not see a delay in the generalization w.r.t. training accuracy/loss in the recognition model with hardtanh despite of an initial phase without compression in two layers. Further learning experiments, including a second run of this last experiment, are presented in the Supplementary Material [12].

## 4 Conclusion and perspectives

We have presented a class of deep learning models together with a tractable method to compute entropy and mutual information between layers. This, we believe, offers a promising framework for further investigations, and to this aim we provide Python packages that facilitate both the computation of mutual informations and the training, for an arbitrary implementation of the model. In the future, allowing for biases by extending the proposed formula would improve the fitting power of the considered neural network models.

We observe in our high-dimensional experiments that compression can happen during learning, even when using ReLU activations. While we did not observe a clear link between generalization and compression in our setting, there are many directions to be further explored within the models presented in Section 2. Studying the entropic effect of regularizers is a natural step to formulate an entropic interpretation to generalization. Furthermore, while our experiments focused on the supervised learning, the replica formula derived for multi-layer models is general and can be applied in unsupervised contexts, for instance in the theory of VAEs. On the rigorous side, the greater perspective remains proving the replica formula in the general case of multi-layer models, and further

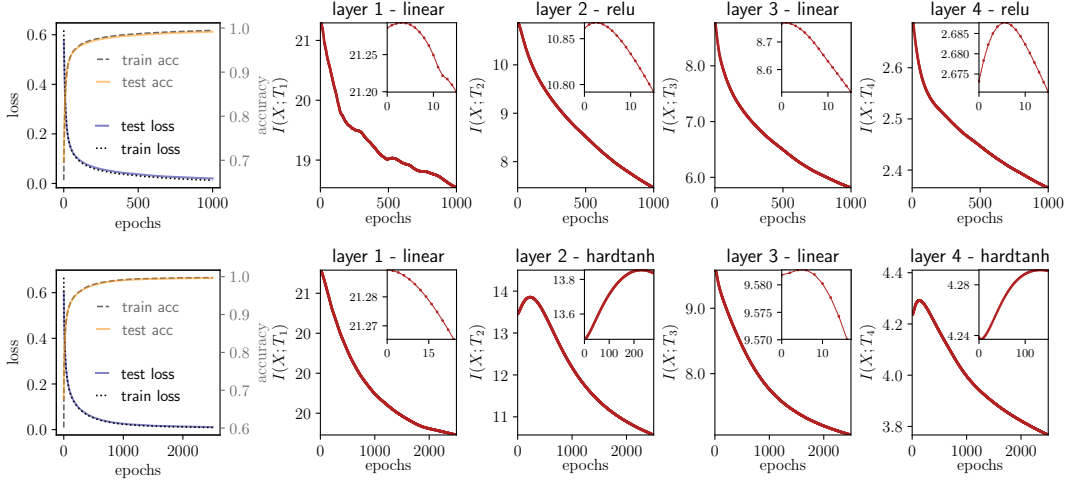

Figure 4: Training of two recognition models on a binary classification task with correlated input data and either ReLU (top) or hardtanh (bottom) non-linearities. Left: training and generalization cross-entropy loss (left axis) and accuracies (right axis) during learning. Best training-testing accuracies are 0.995 - 0.991 for ReLU version (top row) and 0.998 - 0.996 for hardtanh version (bottom row). Remaining colums: mutual information between the input and successive hidden layers. Insets zoom on the first epochs.

confirm that the replica formula stays true after the learning of the USV-layers. Another question worth of future investigation is whether the replica method can be used to describe not only entropies and mutual informations for learned USV-layers, but also the optimal learning of the weights itself.

## Acknowledgments

The authors would like to thank Léon Bottou, Antoine Maillard, Marc Mézard, Léo Miolane, and Galen Reeves for insightful discussions. This work has been supported by the ERC under the European Union's FP7 Grant Agreement 307087-SPARCS and the European Union's Horizon 2020 Research and Innovation Program 714608-SMiLe, as well as by the French Agence Nationale de la Recherche under grant ANR-17-CE23-0023-01 PAIL. Additional funding is acknowledged by MG from "Chaire de recherche sur les modèles et sciences des données", Fondation CFM pour la Recherche-ENS; by AM from Labex DigiCosme; and by CL from the Swiss National Science Foundation under grant 200021E-175541. We gratefully acknowledge the support of NVIDIA Corporation with the donation of the Titan Xp GPU used for this research.

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
