[Reviews · NeurIPS 2018]

Reviewer 1



This work theoretically analyzed entropies and mutual information in a wide class of fully-connected deep neural networks. In details, it derived a novel analytical estimator of entropies (Claim 1) based on the replica method under certain random parameterization and high dimensional limit. A rigorous proof is also given in two-layered networks by using the adaptive interpolation method. The Author(s) confirmed the validity of the theory in various experiments. Especially, it coincided well with the exact solutions of mutual information in deep linear networks. The studies of information bottleneck have ever been limited to small networks or simplified linear ones [Shwartz-Ziv&Tishby 2017, Saxe et al. ICLR2018]. This work highly contributes to give a novel and helpful insight into them and suggests that the compression of the information happens even in ReLU networks but it will be uncorrelated to generalization capacity. Both main text and supplementary material are well organized and their claim seems to be clear enough. Therefore, I recommend this paper for acceptance without any major revision. If possible, It would be helpful to give comments on the following things. * It would be better to generalize your results to models with bias parameters. It will make your theory more easy to use in practice, although it will make your estimator a bit messier. * I am wondering on the optimization properties of the estimator (2) in Claim 1. In Supplementary Materials, Author(s) proposed two types of algorithms (fixed-point iteration and ML-VAMP state evolution). Are there any sub-optimal solutions for minimizing the potential \phi_l when you use these algorithms? Typo in Line 237: samples,they -> samples,_(space) they

Reviewer 2



This paper add to an ongoing discussion about the information bottleneck theory of deep learning. The main contributions of this submission is a theory of the entropy and the mutual information in deep networks that is derived leveraging the replica trick in statistical physics. These results are corroborated by careful experiments. Finally, an application to investigate the contentious compressive phase of learning is performed. Overall the exposition is exceptionally clear and whatever ones opinion about the information bottleneck theory, tools to gain theoretical insight into the mutual information seem like they ought to be extremely valuable. Overall, it is also great to see an application of the replica trick in the context of deep learning. As an aside, there is a lot going on in this paper and I think there really isn't sufficient time to do a proper job of going through the details of the theory. I therefore list my overall confidence as low. I look forward to taking some time to do so in the future.

Reviewer 3



Contributions of the paper: The authors consider a stylized statistical model for data that respects neural network architecture, i.e. a Markov structure of the type T_\ell = \varphi(W_\ell*T_{\ell-1}, \xi_\ell ) where T_0 = X is the input, T_L = y is the output label, $W_\ell$ are random, independent weight matrices, \varphi is a nonlinearity applied elementwise on its first argument, possibly using external randomness \xi_\ell. For data generated from this specific model, they make the following contributions. 1. They show that under this stylized model, one can obtain a simple formula for the (normalized i.e. per unit) entropy H(T_\ell)/n and mutual information I(T_\ell; X)/n between the input data and each successive layer, in the high-dimensional limit. This formula is, in general, derived using the non-rigorous replica method from statistical physics. 2. For a two-layer network and a class of nonlinearities, they rigorously establish this formula, using a technique of `adaptive' interpolation. 3. Based on this model, they provide some experimental results for two synthetic scenarios: the first based on the teacher-student model, and the second for generative models, such as variational autoencoders. The experimental results are multi-faceted and include a comparison with entropy/mutual information estimators, validation of the replica formula, and some applications to the recent information bottleneck proposal of Tishby et al. Summary: The paper is a solid contribution, and I would argue that it is a clear accept. It would make a good addition to the NIPS program and is of potential interest to communities possibly unfamiliar with the approaches in the paper. Comments for the authors: There are multiple aims that you have in the experiments section: which point to interesting applications for your results, like comparing entropy estimators, validating the bottleneck theory, potential training innovations. Further, some of your experiments aim to validate the replica formula where it is not known to hold, e.g. during training (thanks to dependence). It would be good to add an experiments summary that clarifies these aims and points the readers to the relevant sections. You might have to move some of the discussion due to space constraint. I have read the author response. My scores are unchanged, I look forward to the final version of the article.